# Association of Barriers, Fear of Falling and Fatigue with Objectively Measured Physical Activity and Sedentary Behavior in Chronic Stroke

**DOI:** 10.3390/jcm10061320

**Published:** 2021-03-23

**Authors:** M. Luz Sánchez-Sánchez, Anna Arnal-Gómez, Sara Cortes-Amador, Sofía Pérez-Alenda, Juan J. Carrasco, Assumpta Climent-Toledo, Gemma Victoria Espí-López, Maria-Arantzazu Ruescas-Nicolau

**Affiliations:** 1Physiotherapy in Motion, Multispeciality Research Group (PTinMOTION), Department of Physiotherapy, University of Valencia, 46010 Valencia, Spain; M.Luz.Sanchez@uv.es (M.L.S.-S.); Sofia.Perez-Alenda@uv.es (S.P.-A.); Juan.J.Carrasco@uv.es (J.J.C.); asclito@alumni.uv.es (A.C.-T.); Arancha.Ruescas@uv.es (M.-A.R.-N.); 2Research Unit in Clinical biomechanics—UBIC, Department of Physiotherapy, University of Valencia, 46010 Valencia, Spain; Sara.Cortes@uv.es; 3Intelligent Data Analysis Laboratory, University of Valencia, 46100 Burjassot, Spain; 4Exercise Intervention for Health (EXINH), Department of Physiotherapy, University of Valencia, 46010 Valencia, Spain; Gemma.Espi@uv.es

**Keywords:** stroke, sedentary behavior, physical activity, barriers, fear of falling, fatigue, accelerometer

## Abstract

Understanding the fostering factors of physical activity (PA) and sedentary behavior (SB) in post-stroke chronic survivors is critical to address preventive and health interventions. This cross-sectional study aimed to analyze the association of barriers to PA, fear of falling and severity of fatigue encountered by stroke chronic survivors with device-measured PA and SB. Ambulatory community-dwelling post-stroke subjects (≥six months from stroke onset) were evaluated and answered the Barriers to Physical Activity after Stroke Scale (BAPAS), Short Falls Efficacy Scale-International (Short FES-I) and Fatigue Severity Scale (FSS). SB and PA were measured with an Actigraph GT3X+ accelerometer for ≥seven consecutive days. Stepwise multiple linear regression analysis was employed to identify factors associated with PA and SB. Fifty-seven participants (58.2 ± 11.1 years, 37 men) met the accelerometer wear–time criteria (three days, ≥eight h/day). The physical BAPAS score explained 28.7% of the variance of the prolonged sedentary time (β = 0.547; *p* < 0.001). Additionally, the walking speed (β = 0.452) together with physical BAPAS (β = −0.319) explained 37.9% of the moderate-to-vigorous PA time (*p* < 0.001). In chronic post-stroke survivors, not only the walking speed but, also, the perceived physical barriers to PA are accounted for the SB and PA. Interventions to reverse SB and to involve subjects post-stroke in higher levels of PA should consider these factors.

## 1. Introduction

By 2025, it is expected that 1.5 million European people will suffer a stroke each year due to aging of the population and the rising incidence rate observed in young adults [1]. Despite recent advances in the acute-phase management of stroke survivors, disability rates remain high [2]. Thereby, with an estimated 27% increase in the number of people surviving a stroke in Europe, the absolute burden of stroke is expected to continue to increase over the next 30 years [3]. Healthy behaviors and preventive practices might help manage chronic health conditions successfully [4]. In this sense, recent evidence strongly supports the benefits of physical activity (PA) for stroke survivors [5], understanding PA as body movement produced by the contraction of skeletal muscles, thus increasing energy expenditure [6]. To reduce stroke risk factors, the American Heart and Stroke Associations recommends post-stroke survivors who are capable of engaging in PA to at least perform three to four 40-min sessions per week of moderate-to-vigorous-intensity aerobic exercise [7]. However, physical inactivity is quite common post-stroke, leading to physical deconditioning [8,9].

Regardless of PA, sedentary behavior (SB) (defined as any waking behavior characterized by low energy expenditure (≤1.5 metabolic equivalents) while in a sitting, reclining or lying posture) [10] negatively affects cardiovascular health [9], particularly when sedentary time is accumulated in long, uninterrupted periods [11]. It should be noted that chronic stroke survivors are more sedentary and less active than age-matched controls, with independently ambulatory subjects spending 75% of their waking hours sitting down each day and showing very low PA levels [12].

Considering that inadequate levels of PA of at least moderate intensity are a significant risk factor for recurrent stroke and that sitting time is an independent risk factor for cardiovascular disease-related mortality [9,13], public health programs and policies in chronic stroke should prevent physical inactivity and SB [4]. In this sense, to change these behaviors, it is critical to know the modifiable factors associated with them in people with sequelae of chronic stroke [8,14].

In the last years, the growing emerging literature in this field [15,16,17,18,19,20] shows that both physical inactivity and SB post-stroke are an increasing health concern and focus of research [21]. However, the evidence about modifiable factors influencing objectively measured PA and SB is scarce in the chronic phase of stroke [22,23,24]. An exploratory study found that walking speed and self-reported physical function may influence the amount of time spent of, at least, moderate-intensity PA, but much of the variance in daily sitting time remains unexplained [23]. In this sense, relevant factors, including barriers to PA, fear of falling and fatigue, may also have an influence in SB and physical inactivity, but their association has not yet been well-established in chronic stroke survivors. Moreover, no previous study has focused on the association of these modifiable factors with accelerometer-based PA and SB in this population.

Based on the existing evidence, it was hypothesized that barriers, fear of falling and fatigue would show significant associations with PA and SB. Thus, the aim of this cross-sectional study was to analyze the association of perceived barriers to PA, fear of falling and severity of fatigue with PA and SB in chronic stroke survivors.

## 2. Materials and Methods

### 2.1. Study Design

This study used a cross-sectional observational design. The study conformed to the Declaration of Helsinki [25] and was approved by the University Research Ethics Committee (No. 1563377228465). All participants were fully informed about the study’s purpose and procedures and provided written informed consent. This article adheres to the STROBE guidelines [26].

### 2.2. Participants

Adults (≥18 years old) with sequelae after stroke of at least 6 months of evolution were included. Participants were eligible for inclusion if they had been living at home for at least 2 months since their most recent stroke and had residual gait deficits but were able to walk independently at home with or without walking aids (Functional Ambulation Classification of the Hospital at Sagunto ≥ 2) [27]. Participants also had the ability to understand verbal instructions to undergo the assessment tests. People who had a poor vital prognosis or who suffered from other pathologies or disorders that may alter the development of the study (e.g., blindness, dementia, musculoskeletal or cardiovascular conditions that contraindicate the performance of PA, etc.) were excluded. Participants were recruited from brain injury associations, physiotherapy outpatient services and community exercise classes adapted to neurological impairments located in the region of Valencia (Spain).

### 2.3. Procedures and Outcomes

All participants undertook a face-to-face assessment at the University of Valencia laboratories or in their usual care centers. This assessment was followed by 7 days of objective activity monitoring. During the evaluation (Appendix A and Appendix A), demographic characteristics were gathered from interviews with the participants, who were also asked if they lived alone or accompanied. Clinical information (history of stroke and comorbidities) was obtained from medical records, and the Charlson Comorbidity Index was used to classify comorbid conditions [28]. In the original scale, the maximum possible score was 30 points. However, in this study, as in previous research, cerebrovascular disease and hemiplegia were not considered comorbidities, yielding a maximum score of 27 points, which indicated a high level of comorbidity [29].

Then, cognitive ability and functional capacity were assessed with the Montreal Cognitive Assessment (MoCA) [30] and the modified Rankin Scale (mRS) [31], respectively. Regarding the latter, most stroke trials define a favorable functional outcome as mRS grade ≤2 [32]. Therefore, we rated participants who scored ≤2 as “No disability or slight” and those who scored 3 to 5 as “Moderate or severe disability”. The Patient Health Questionnaire-9 (PHQ-9) was employed for measuring severity of depression symptoms [33].

Additionally, the Stroke Impact Scale-16 (SIS-16) was employed to assess self-reported physical function [34]. The SIS-16 is an interview-administered questionnaire focused on the quality-of-life levels related to physical function. Thus, subjects were asked to rank in a Likert-type scale (5 = Not difficult at all to 1 = inability to complete the item) the difficulty they found during the last 2 weeks when performing 16 skills related to 4 physical domains (i.e., strength, hand function, mobility and activities of daily life). The scores were transformed into a 0–100 scale. A higher score indicated better levels of subjective health-related quality of life. The SIS-16 has shown good discriminant validity in stroke subjects, excellent internal consistency (Cronbach’s alpha = 0.94) and test–retest reliability (ICC = 0.95) [35].

Afterward, height and weight were objectively assessed using a portable stadiometer and a weighing machine. Then, the body mass index (BMI) was calculated (kg/m^2^). Additionally, the comfortable walking speed was measured with the 10-Meter Walk Test (10 mWT). Participants were timed while walking at self-selected pace along a 10-m corridor. Distances were provided at the beginning and end of the walkway to allow participants to accelerate/decelerate, so the intermediate 6 m were timed. Walking speed was determined with the equation 6 m/s needed to walk the 6 m. The measurement was carried out twice, and the average of the two trials was used in the statistical analyses. This test has demonstrated excellent reliability in subjects with chronic stroke (ICC = 0.94), with no discernible differences between different tests [36].

Finally, participants completed the following outcomes measures:Barriers to physical activity

To assess participant’s barriers to PA, the Barriers to Physical Activity after Stroke Scale (BAPAS) [37] was used. This self-reported questionnaire contains 14 items: 7 explore behavioral dimensions such as fatigue/mood and motivation; thus, they are related to behavioral barriers (BAPAS-behav); the other 7 explore locomotor problems and comorbidities, so they are related to physical barriers (BAPAS-physic). The items are scored on a 6-point Likert scale as follows: Strongly agree (5), agree (4), slightly agree (3), slightly disagree (2), disagree (1) and strongly disagree (0). Consequently, the scores range from 0 to 70 (maximal sub-scores of 35 each), and higher scores indicate higher barriers to PA. The BAPAS has shown good internal consistency (Cronbach’s alpha = 0.86) and test–retest reliability (ICC = 0.91 BAPAS; ICC = 0.92 BAPAS-behav; ICC = 0.91 BAPAS-physic) for stroke survivors with low-to-moderate disability [37].

2.Fear of falling

To assess the fear of falling, the Short Falls Efficacy Scale-International (Short FES-I) was used. This index indicates the level of perceived confidence an individual has carrying out activities of daily living without falling. The short version contains 7 items scored from 1 (not at all concerned) to 4 (very concerned). Therefore, the total score ranges from 7 to 28. The higher the score the more concern the patient has about falling. The internal consistency and test–retest reliability of the Short FES-I are excellent (Cronbach’s alpha = 0.92, ICC = 0.83) for elderly people [38]. The Spanish version of the Short FES-I showed good reliability and validity in elderly people [39].

3.Severity of fatigue

The severity of fatigue and its effect on a person’s activities and lifestyle were assessed with the Fatigue Severity Scale (FSS), which is the most widely used measure of fatigue in people with stroke [40]. It is a 9-item self-reporting questionnaire measuring the severity of fatigue in different situations during the past week. The level of agreement with each item is ranked from 1 (strong disagreement) to 7 (strong agreement), and the final score results from averaging the scores of the 9 items. The presence of fatigue is considered when a FSS score ≥ 4, while higher scores indicate higher levels of fatigue. FSS has demonstrated excellent internal consistency in subjects with stroke (Cronbach’s alpha = 0.928) and exhibits good test–retest reliability (ICC = 0.742) [41].

4.Activity Monitoring Measurement

Habitual PA was monitored at 30 Hz using triaxial accelerometers (Model Actigraph wGT3X-BT; Pensacola, FL, USA) [42]. Participants were instructed to wear the accelerometer for ≥7 consecutive days on an elasticized band positioned over the Anterior Superior Iliac Spine of their nonparetic hip. Participants were apprised to only remove the device when showering (or during water-based activities) and at night. They were also instructed to log the times the accelerometer was worn or removed each day in a diary. PA data were downloaded and processed with ActiLife 6 software, and episodes of ≥90 min of consecutive zeroes were considered non-wear time and not included in the analysis [43]. Participants with ≥3 days of valid PA data (i.e., ≥8 h of wear time) were included in the analysis [44]. We used the accelerometer count cut-points established by Freedson et al. [45], which define sedentary time as 99 counts per minute (cpm) or less, light PA (LPA) as between 100 and 1951 cpm and moderate-to-vigorous PA (MVPA) as 1952 cpm or greater. This approach is consistent with previous studies of PA in stroke subjects [12,23].

For statistical analysis, the data recorded was averaged per day (i.e., considering only the number of days that contained valid measurements). Then, all outcome measures were adjusted for wear time. Thus, our outcome measures were:Sedentary time, percentage of time in SB of the total wear time (time in SB/Total wear time*100).Prolonged sedentary time, percentage of time in prolonged SB (total time in SB bouts ≥ 30 min) of the total wear time (time in prolonged SB/Total wear time*100).LPA, percentage of time spent in LPA of the total wear time (time in LPA/Total wear time*100).MVPA, percentage of time spent in MVPA of the total wear time (time in MVPA/Total wear time*100).

### 2.4. Sample Size Calculation

A previous study [23] reported an independent association (R^2^ = 0.282) between walking speed, the SIS physical domain scores, stroke severity and level of independence and prolonged sitting time in stroke subjects. We therefore set the expected R^2^ of the multiple regression analysis in this study at 0.282. The effect size (f^2^) was calculated by the equation R^2^/(1 − R^2^) [46]. An a priori sample size calculation with a f^2^ of 0.39, power of 0.80, alpha error of 0.05 and 9 to 10 predictor variables indicated that a sample size of at least 49–52 participants would be required. Adding a loss percentage of 15%, a sample size of at least 57 to 60 participants would be required. The sample size calculation was conducted using G*Power version 3.1.9.4 (Heinrich-Heine-Universität, Düsseldorf, Germany) [47].

### 2.5. Data Analysis

The normality of continuous data was verified using the Kolmogorov–Smirnov test. Descriptive measures are shown as Mean (Sd), Median (25th–75th percentiles) or frequency (percentage), as appropriate. Correlation analyses were used as the initial step to determine the variables to be included in the regression analysis. The association of the continuous variables with sedentary behavior and PA was assessed using Pearson’s or Spearman’s correlation coefficients, as appropriate. Correlation coefficients were interpret as small 0.1–0.3, medium 0.3–0.5 and large 0.5–1.0 [48]. The association of sedentary behavior and PA with the dichotomous variables (living arrangement and mRS) were assessed using a *t*-test for independent samples or the Mann–Whitney *U* test. Stepwise multiple regression analyses were used to identify the factors that were independently associated with sedentary behavior and PA variables. For sedentary time and prolonged sedentary time, the Charlson Comorbidity Index, walking speed, SIS-16, Short FES-I, BAPAS-physic and mRS were set as the independent variables. For LPA and MVPA, the same independent variables and FSS were included in the model as the independent variables. All regression analyses were adjusted for potential confounders (age, sex and BMI).

Assumptions for linear regression were checked: Homoscedasticity was tested by plotting the residuals versus the fitted values, the presence of multicollinearity was determined by a variance inflation factor (VIF) larger than 3 and the influential points were inspected with Cook’s distance.

Statistical significance was set at *p* < 0.05. However, to correct for multiple testing in the regression models, the significance level was set at a <0.05/4 = 0.0125. All analyses were carried out using IBM SPSS Statistics software (Version 26.0; IBM Corp, Armonk, NY, USA).

## 3. Results

### 3.1. Participants’ Characteristics

Sixty-one people (58.1 ± 10.8 years, 39 men) with stroke were recruited. Four participants did not meet accelerometer wear time criteria (Figure 1). Therefore, the data from fifty-seven participants (58.2 ± 11.1 years, 37 men) were analyzed. Table 1 presents the demographic and clinical characteristics of the sample. On average, participants showed overweight (BMI = 28.6 ± 4.6), mild cognitive impairment (MoCA score = 23.0 (20.0; 25.5)) and mild depressive symptoms (PHQ9 score = 5.0 (2.0; 8.0)). Twenty-one (36.8%) presented moderate or severe disability, and only six (10.5%) lived alone.

### 3.2. Results of the Barriers, Fear of Falling and Fatigue Scales and Activity Monitoring Measurement

As Table 2 shows, 75% of the participants reported a score of less than 30 points out of 70 on the BAPAS scale. Subjects reported higher physical (12.0 (5.0; 17.0)) than behavioral barriers (7.0 (4.0; 12.0)), and two subjects reported not having any barrier. Participants scored higher for the barriers “I have a loss of muscle strength (paralysis)” (2.3 ± 2.0) and “I have spasticity (muscle stiffness)” (2.3 ± 2.1). Meanwhile, in our sample, the lowest scored barrier was “I lack financial resources” (0.7 ± 1.5).

Table 2 also shows the results of the fear of falling (Short FES-I score) and severity of the fatigue (FSS score). A total of 75% of participants reported a score of less than 12 points out of 28 on the Short FES-I. Moreover, 37 participants (64.9%) did not show fatigue.

Participants spent a mean of 636.1 ± 119.2 sedentary minutes per day, which is 81.1% ± 11.8% of the accelerometer wear time per day. Meanwhile, they only spent a median of 5.80 (2.45; 22.40) minutes in MVPA, which is the 0.87% (0.41; 2.53) of the accelerometer wear time per day.

### 3.3. Association between Barriers, Fear of Falling, Fatigue and Level of Physical Activity, and Sedentary Behavior

In Table 3, the results of the correlation analyses and the differences in dichotomous variables of factors influencing SB and PA are presented. Mainly, the BAPAS scores, especially the BAPAS-physic scores, showed the highest correlation coefficients with the prolonged sedentary time, which was of a moderate degree in both cases (ρ = 0.451, *p* < 0.001 and ρ = 0.582, *p* < 0.001, respectively). Likewise, the walking speed, BAPAS-physic scores, SIS-16 scores and BAPAS scores showed the strongest association with MVPA, resulting in moderate correlation coefficients (ρ = 0.592, *p* < 0.001, ρ = −0.578, *p* < 0.001, ρ = 0.497, *p* < 0.001 and ρ = −0.471, *p* < 0.001, respectively). The item “I have other medical conditions” of the BAPAS-physic score showed the highest correlation coefficient with prolonged sedentary time (0.535; *p* < 0.001), and the item “I have spasticity, muscle stiffness” was inversely correlated with MVPA, showing the highest correlation coefficient (−0.601; *p* < 0.001) (Appendix A). Moreover, participants who were moderate or severely disabled spent significantly more time sedentary (85.48% ± 11.10%) per day than those slightly or not disabled (78.48% ± 11.53%, t = −2.241, *p* = 0.029). Accordingly, participants having a slight or no disability spent significantly more time in MVPA (1.48%) per day than those moderate or severe disabled (0.52%, *U* = 257.000, *p* = 0.045), although both percentages of time were very small.

As Table 3 shows, the Charlson Comorbidity Index, walking speed, SIS-16, Short FES-I, BAPAS, BAPAS-physic and mRS were the factors that significantly influenced the SB. For PA, the same variables together with FSS had statistically significant effects. Thus, all of them, except BAPAS, were entered into the stepwise multiple linear regression analyses. Both BAPAS and BAPAS-physic showed a significant correlation with our dependent variables and between them. Therefore, to avoid multicollinearity, we selected BAPAS-physic, as it showed higher correlation coefficients.

The results of the stepwise multiple linear regression analyses are shown in Table 4. None of the seven independent variables entered into the regression were retained in the model to explain the variance in the LPA. On the other hand, the regression analysis revealed that, out of the six independent variables, only the walking speed was retained in the model and explained 13.5% (β = −0.388; *p* = 0.003) of the variance in the sedentary time. The walking speed was negatively associated with the sedentary time. In the case of prolonged sedentary time, only the BAPAS-physic score was retained in the model, explaining 28.7% (β = 0.547; *p* < 0.001) of this variable variance. The BAPAS-physic score was positively associated with the daily prolonged sedentary time. Finally, in the regression analysis involving MVPA, of the seven independent variables, only the walking speed and the BAPAS-physic score were retained in the model. Together, they explained 37.9% of the variance in MVPA. The walking speed explained 29.8% (β = 0.452; *p* < 0.001) of the daily time spent on MVPA. By adding the BAPAS-physic score, the explained variance increased to 37.9% (β = −0.319; *p* < 0.001). The walking speed was positively associated with daily MVPA, whereas the BAPAS-physic score was negatively associated.

## 4. Discussion

The present study showed that, for our sample, perceived barriers were significantly associated with PA and SB. Specifically, the BAPAS-physic score was positively associated with daily prolonged sedentary time, explaining 28.7% (F = 23.495; *p* < 0.001) of its variance. This indicated that participants who perceived higher physical barriers to PA were more likely to spend daily prolonged time in SB. Moreover, BAPAS-physic score together with the walking speed explained 37.9% of the variance in MVPA. The walking speed was positively associated with the daily MVPA, whereas the BAPAS-physic score was negatively associated. This indicated that participants who had a faster walking speed and perceived fewer physical barriers to PA were more likely to spend more daily time in MVPA.

These results reinforce the recent literature that indicated that barriers to PA are likely to be critical factors in the regular practice of PA and its maintenance over time [37]. Several authors have focused on the presence of perceived barriers to PA [8,49,50] or to exercise [51,52,53] in post-stroke participants. However, none of them studied their association with objectively measured PA and SB. Jackson et al. (2018) [8] studied the correlation between barriers and PA, and they found significant relationships between self-reported PA and fear of falling, functional mobility and beliefs relating to PA. In our sample, only 33.3% of the participants referred fear of falling as a barrier to PA, but concurring with Jackson et al. (2018) [8], the physical barriers were scored higher. In particular were those related to functional mobility, such as the loss of muscle strength and muscle stiffness. It should be noted that muscle weakness is a common consequence of stroke and could result in abnormal posture and stretching reflexes and, finally, in a loss of voluntary movement [54]. Therefore, based on our results and according to the stroke guidelines, strength training is highly recommended in rehabilitation post-stroke [55]. Thus, this type of intervention not only can improve functionality but, also, may diminish the perceived physical barriers to PA.

As modifiable factors, and taking into account their associations with time spent in prolonged SB and in MVPA, perceived barriers should be considered in future interventions of stroke survivors to reduce the stroke risk factors. The participants in our study showed higher physical than behavioral barriers. In this line, Drigny et al. (2019) [37] suggested an adaptive supervised PA intervention to improve PA practice post-stroke, while an adaptive self-managed PA intervention would be more appropriate when behavioral barriers were higher.

While previous studies have indicated that PA and SB in stroke survivors could be influenced by fear of falling and fatigue [18,19,29,56,57,58,59], these were not significantly predictive factors of PA and SB in our sample. However, the majority of our participants (*n* = 37, 64.9%) had scores under the cut-off value for fatigue on the FSS [40], and the fear of falling on the Short FES-I in this group was relatively low.

In line with the current literature [23,60], on average, our sample was largely sedentary. The participants spent 636.08 min of sedentary time per day and 16.63 min of MVPA. Although 16.63 min on average in MVPA doubled the time spent in MVPA reported by other authors [23,60], only nine participants (15.80%) achieved the recommendation of an average of 30 min of daily MVPA. This percentage is far from the 75% of subjects who fulfilled this recommendation in the study carried out by Fini et al. (2019) [15]. Moreover, participants in the study carried out by Wondergem et al. (2019) [17] spent, on average, 36 min per day in MVPA. Thus, our sample was not only sedentary but, also, physically inactive. On the other hand, in the present study, none of the independent factors studied were retained in the regression model to explain the variance in LPA. It should be noted that Gothe et al. (2019) [60] pointed out the relevance of this level of PA, because they showed that LPA, not MVPA, significantly predicted the self-reported function. Thus, knowing the difficulties to reach MVPA after a stroke, they suggested that those daily routines that keep stroke survivors physically engaged in lighter tasks (e.g., leisurely walks or nonstrenuous household chores) also contribute to their better physical functioning. Then, future research should study which modifiable factors influence LPA post-stroke and include interventions to increase LPA post-stroke. On average, our participants spent 136.82 min per day in LPA, which was lower than the 203.29 min presented by American post-stroke subjects in the study of Gothe et al. (2019) [60] but higher than the 106 presented by African post-stroke subjects in the study of Joseph et al. (2018) [24]. Although, the different cut-off points used for intensity categories could have influenced these differences.

In chronic stroke survivors, previous exploratory studies have found greater self-reported physical disability and slower walking speed to be associated with more time spent sitting (in particular in prolonged sitting bouts), as well as less time in MVPA [23], and age, severity of stroke and not attending outpatient rehabilitation were associated with sedentary activity in their sample [24]. In the present study, these variables were included, except attendance to outpatient rehabilitation, since all the participants attended. In addition, our variables of interest were included in the analysis. Our results support walking speed as an independent factor for SB. In our model, only walking speed was retained and explained 13.5% (F = 9.744; *p* = 0.003) of the variance in sedentary time. The walking speed was negatively correlated with sedentary time. Thus, participants who had lower walking speed were more likely to spend more time in SB. Moreover, our results also support walking speed as an independent factor for MVPA. As explained previously, walking speed together with BAPAS-physic score explained 37.9% of the variance in MVPA in our sample. The current literature points out walking speed as an important modifiable factor associated to SB and PA [15,44]. In this sense, our research also highlights the relevance of the physical perceived barriers in SB and MVPA, because in our sample, this factor was retained in the regression model, whereas the other factors previously indicated did not explained the variance in the SB and PA.

Our study is limited by its cross-sectional design, which did not allow us to conclude on the causality between independent factors and PA or SB. Likewise, recruitment via attendance at associations and specialized centers could impose some type of bias, although this is unlikely, given that attending these centers is quite common in Spain in the chronic stage of the stroke and that the results of the PA levels did not differ from previous studies. Acceptance to participate in the study may also involve another form of bias. Finally, since we studied chronic stroke survivors, our results cannot be generalized to acute or subacute stroke. Since subjects post-stroke tend to overestimate PA and underestimate SB [29], the major strength of our study is the objective measurement of PA and SB without possible bias from the subjective character of a questionnaire. Moreover, to our knowledge, no published studies with device-measured PA and SB data have investigated the association of perceived barriers, fear of falling and fatigue with PA and SB in chronic subjects post-stroke.

## 5. Conclusions

The perceived barriers were associated with objectively measured PA and SB in our sample of chronic post-stroke survivors. Specifically, the perceived physical barriers were positively associated with daily prolonged sedentary time and negatively associated with daily MVPA. Moreover, the walking speed was positively associated with daily MVPA. No associations were found for fear of falling and severity of fatigue. Therefore, not only the walking speed but, also, the perceived physical barriers to PA account for the SB and PA for this population. Health interventions aimed to reverse the SB and to involve post-stroke subjects in higher levels of PA should approach these factors.

## Figures and Tables

**Figure 1 jcm-10-01320-f001:**
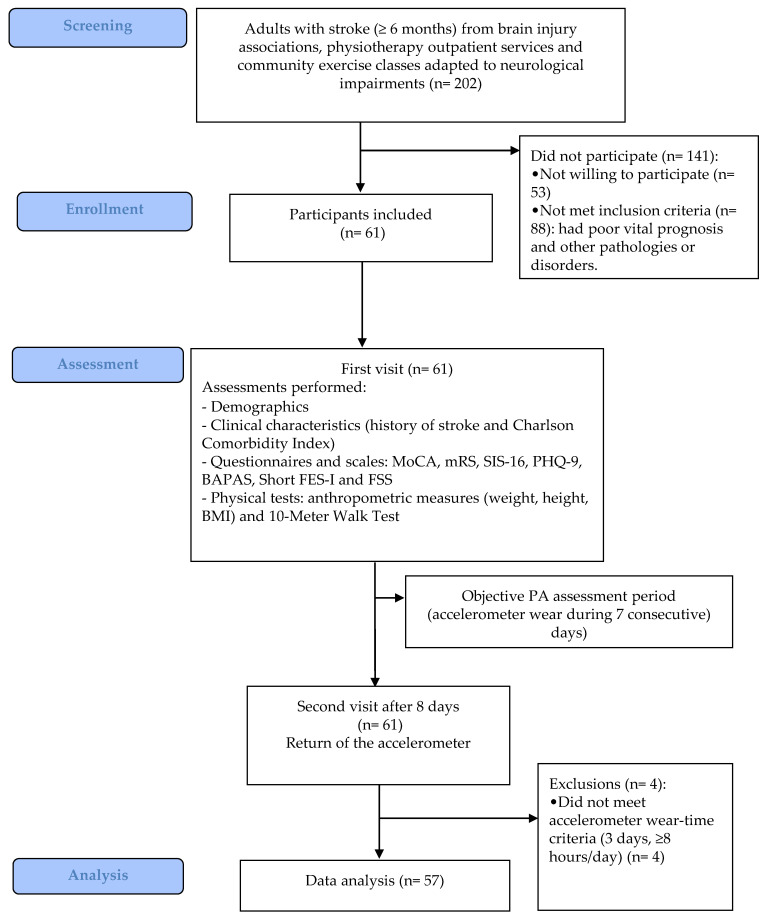
Flow diagram of study participation. MoCA: Montreal Cognitive Assessment, mRS: Modified Rankin Scale, SIS-16: Stroke Impact Scale-16, PHQ-9: Patient Health Questionnaire-9, BAPAS: Barriers to Physical Activity after Stroke Scale, Short FES-I: Short Falls Efficacy Scale-International, FSS: Fatigue Severity Scale, BMI: Body mass index and PA: Physical activity.

**Table 1 jcm-10-01320-t001:** Participants ‘demographic and clinical characteristics (*n* = 57).

		Mean ± Sd, Median [IQR] or *n* (%)
Age (years)		58.2 ± 11.1
Sex, males, *n* (%)		37 (64.9)
Weight (kg)		80.5 ± 14.5
Height (m)		1.7 ± 0.1
Body Mass Index (kg/m^2^)		28.6 ± 4.6
Stroke type	Ischemic	35 (61.4)
	Hemorrhagic	22 (38.6)
Time since stroke onset (months)		64.0 [38.5; 105.5]
Paretic side	Right	23 (40.4)
	Left	33 (57.9)
	Both	1 (1.8)
Living arrangement	Alone	6 (10.5)
	Accompanied	51 (89.5)
Charlson Comorbidity Index (score)		0.0 [0.0; 1.0]
MoCA, score (0–30)		23.0 [20.0; 25.5]
mRS	No disability or slight (%)	36 (63.2)
	Moderate or severe disability (%)	21 (36.8)
PHQ9, score (0–27)		5.0 [2.0; 8.0]
SIS-16, score (0–100)		85.9 [73.4; 92.2]
Walking speed (m/s)		0.8 ± 0.4

Sd: Standard deviation, IQR: Interquartile range (25th–75th percentiles), MoCA: Montreal Cognitive Assessment, mRS: Modified Rankin Scale, PHQ-9: The Patient Health Questionnaire-9 and SIS-16: The Stroke Impact Scale-16.

**Table 2 jcm-10-01320-t002:** Results of the barriers, fear of falling and fatigue scales and the activity monitoring measurement (*n* = 57).

	Mean ± Sd or Median [IQR]
BAPAS, score (0–70)	20.0 [10.0; 30.0]
BAPAS-physic, score (0–35)	12.0 [5.0; 17.0]
BAPAS-behav, score (0–35)	7.0 [4.0; 12.0]
Short FES-I, score (7–28)	10.0 [8.0; 12.0]
FSS, score (1–7)	3.5 ± 1.7
No. Accelerometer wear days	6.0 [5.0; 7.0]
Accelerometer wear time (min/day)	789.5 ± 125.0
Sedentary time (min/day)	636.1 ± 119.2
Sedentary time (%)	81.1 ± 11.8
Prolonged sedentary time (min/day)	163.80 [68.65; 263.30]
Prolonged sedentary time (%)	21.32 [9.53; 35.86]
LPA (min/day)	112.90 [64.05; 192.30]
LPA (%)	13.97 [8.22; 25.36]
MVPA (min/day)	5.80 [2.45; 22.40]
MVPA (%)	0.87 [0.41; 2.53]

Sd: Standard deviation, IQR: Interquartile range (25th–75th percentile), BAPAS: Barriers to Physical Activity after Stroke Scale, BAPAS-physic: Physical barriers from BAPAS, BAPAS-behav: Behavioral barriers from BAPAS, Short FES-I: Short Falls Efficacy Scale-International, FSS: Fatigue Severity Scale, LPA: Light physical activity and MVPA: Moderate-to vigorous physical activity.

**Table 3 jcm-10-01320-t003:** Factors influencing the sedentary behavior and physical activity—correlation analyses and differences in dichotomous variables (*n* = 57).

	Sedentary Time (%)		Prolonged Sedentary Time (%)		LPA (%)		MVPA (%)	
	Coefficient	*p*-Value	Coefficient	*p*-Value	Coefficient	*p*-Value	Coefficient	*p*-Value
CCI	ρ = −0.01	0.942	**ρ = 0.303**	**0.022**	ρ = 0.088	0.515	**ρ = −0.296**	**0.025**
Time since stroke onset	ρ = 0.011	0.934	ρ = −0.154	0.253	ρ = 0.029	0.830	ρ = −0.111	0.410
MoCA	ρ = 0.019	0.887	ρ = 0.004	0.975	ρ = −0.059	0.662	ρ = 0.159	0.238
PHQ9	ρ = 0.023	0.867	ρ = 0.000	0.998	ρ = 0.007	0.961	ρ = −0.065	0.629
Walking speed	**r = −0.388**	**0.003**	**ρ = −0.367**	**0.005**	ρ = 0.257	0.054	**ρ = 0.592**	**<0.001**
SIS−16	ρ = −0.241	0.071	**ρ = −0.420**	**0.001**	ρ = 0.100	0.461	**ρ = 0.497**	**<0.001**
Short FES-I	ρ = 0.096	0.477	**ρ = 0.290**	**0.028**	ρ = 0.010	0.943	**ρ = −0.305**	**0.021**
BAPAS	ρ = −0.073	0.590	**ρ = 0.451**	**<0.001**	ρ = 0.200	0.135	**ρ = −0.471**	**<0.001**
BAPAS_physic	ρ = 0.004	0.979	**ρ = 0.582**	**<0.001**	ρ = 0.145	0.282	**ρ = −0.578**	**<0.001**
BAPAS_behav	ρ = −0.084	0.532	ρ = 0.129	0.337	ρ = 0.141	0.296	ρ = −0.172	0.200
FSS	r = −0.087	0.521	ρ = 0.086	0.525	ρ = 0.134	0.320	**ρ = −0.285**	**0.032**
	**Median [IQR]**	***p*-value**	**Median [IQR]**	***p*-value**	**Median [IQR]**	***p*-value**	**Median [IQR]**	***p*-value**
Living arrangement								
Alone (*n* = 6)	84.02 [72.07;90.76]	0.849 ^b^	18.55 [11.71;33.23]	0.849 ^b^	13.82 [6.79;27.34]	0.829 ^b^	2.01 [0.24;2.57]	0.771 ^b^
Accompanied (*n* = 51)	82.03 [73.58;89.85]		21.62 [9.52;36.05]		13.97 [8.35;25.49]		0.85 [0.43;2.70]	
Modified Rankin Scale (mRS)								
No disability or slight (*n* = 36)	78.05 [72.41;87.64]	**0.029 ^a^**	20.17 [8.00;32.09]	0.118 ^a^	17.52 [9.35;26.20]	0.104 ^a^	1.48 [0.44;3.71]	**0.045 ^b^**
Moderate or severe disability (*n* = 21)	89.02 [78.07;94.67]		27.61 [9.94;41.65]		10.72 [4.89;21.56]		0.52 [0.32;1.12]	

Significant differences are highlighted in bold. ^a^ Student’s *t*-test and ^b^ U Mann–Whitney. BMI: Body Mass Index, LPA: Light physical activity, MVPA: Moderate-to-vigorous physical activity, CCI: Charlson Comorbidity Index, MoCA: Montreal Cognitive Assessment, PHQ9: Patient Health Questionnaire-9, SIS-16: Stroke Impact Scale-16, Short FES-I: Short Falls Efficacy Scale-International, BAPAS: Barriers to Physical Activity after Stroke Scale, BAPAS_physic: Physical barriers from BAPAS, BAPAS_behav: Behavioral barriers from BAPAS, FSS: Fatigue Severity Scale, and IQR: Interquartile range (25th–75th percentile).

**Table 4 jcm-10-01320-t004:** Factors influencing sedentary behavior and physical activity—stepwise multiple linear regression analyses (*n* = 57).

	*B*	95% CI for B	*β*	Adjusted R^2^	SEE
Sedentary time (%)					
Constant	90.781 ± 3.436	83.895 to 97.667	-	-	-
Walking speed	**−12.351 ± 3.957**	**−20.281 to −4.421**	**−0.388**	**0.135**	**10.953**
Prolonged sedentary time (%)					
Constant	11.873 ± 2.850	6.161 to 17.586	-	-	-
BAPAS_physic	**0.941 ± 0.194**	**0.552 to 1.330**	**0.547**	**0.287**	**12.553**
MVPA (%)					
Constant	0.450 ± 1.052	−1.659 to 2.559	-	-	-
Walking speed	**3.780 ± 0.935**	**1.906 to 5.653**	**0.452**	**0.298**	**2.594**
BAPAS_physic	**−0.114 ± 0.040**	**−0.194 to −0.034**	**−0.319**	**0.379**	**2.440**

Significant differences are highlighted in bold. For the sedentary time and prolonged sedentary time, the Charlson Comorbidity Index, walking speed, Stroke Impact Scale-16, Short Falls Efficacy Scale-International, BAPAS-physic and modified Rankin Scale were set as the independent variables. For moderate-to-vigorous physical activity, the same independent variables and the Fatigue Severity Scale were included in the model as the independent variables. *B*: Regression coefficients, followed by the respective standard errors. CI: Confidence interval, β: Standardized regression coefficients, R^2^: Coefficients of determination, SEE: Standard errors of the estimate, BAPAS_physic: Physical barriers from Barriers to Physical Activity after Stroke Scale and MVPA: Moderate-to-vigorous physical activity.

## Data Availability

The data presented in this study are available upon request from the corresponding author.

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
