# Peer review of "Association of Barriers, Fear of Falling and Fatigue with Objectively Measured Physical Activity and Sedentary Behavior in Chronic Stroke"

_jcm, 2021, doi:10.3390/jcm10061320_

Round 1
Reviewer 1 Report
This study investigated the factors influencing the sedentary behavior and physical activity measured objectively using an accelerometer. They showed that perceived barriers and walking speed were independently associated with physical activity and sedentary behavior in chronic post-stroke survivors.
- In Abstract, it would be easier to understand to present the regression coefficient values ​​(beta values) instead of the F values ​​for multivariate linear regression analysis.
- Please clearly present the study design such as study population (inclusion/exclusion), test (type, time point, period), outcomes (type, time point) through flow chart.
- In Table 4, describe exactly in the footnote which independent variables were added.
- BAPAS was divided into physical and behavioral. It is not clear what items are included in behavioral barriers and what items are included in physical barriers. Therefore, it is necessary to present the contents of the questionnaire as a supplementary material. In addition, describe the exact definitions of physical and behavioral barriers in the Methods.
- Among the items of the BAPAS-physic score, indicate which items are most closely related to SB and PA as a figure or supplementary material (ie. physical barriers related to functional mobility such as loss of muscle strength and muscle stiffness).
- For better understanding, it is recommended to present the questionnaire form of Charlson Comorbidity Index, MoCA, PHQ-9, SIS-16, Short FES-I, FSS as supplementary material.
- Like BAPAS, SIS-16 (physical function) is expected to be associated with SB and PA, but it was not. What is the reason?
- Are the factors affecting SB and PA different in subgroup analysis according to stroke type?
- In the Results, the table presents medial(IQR) or regression coefficient values, but the manuscript describes the mean(SD) or F value. Therefore, there may be confusion in the reader's understanding. It is necessary to match the contents of the table and the manuscript.
- In multiple linear regression analysis, enter(method) may be more explanatory than stepwise(method). Is the result similar after analysis even if it is changed to enter(method)? Time since stroke onset also needs to be added as an independent variable.
Reviewer 2 Report
Interesting work. I have some questions for the authors I would appreciate to be answered.
1) from the introduction, I cannot understand what the authors want to do in their research. Are they aiming at descibing better PA and SB? Or rather they are aiming to corroborate PA and SB with insrumental data?
2) Are authors' findings to be cosidered as limited to the chronic stroke cohort? Or would it be possible to assess subacute (by a rehabilitative point of view) individuals?
3) Is Short FES-I validated for post-stroke? Please add reference if available.
4) "These results reinforce recent literature that indicates that barriers to PA are likely to be critical factors in the regular practice of PA and its maintenance over time". So, what is the novelty of this study? what does it add to the literature?
5) "strength training is highly recommended in rehabilitation post-stroke [55], not only to improving functionality but also to diminishing physical barriers". Is this a new finding?
6) Could it be the case that the authors add a section on the generalizability, if any, of their findings?
7) Lastly, I think that the authors should motivate why it is necessary to use instumental assessment to evaluate PA and SB. Maybe the authors yet expressed this concept but I did not understand it. In such a case could the authors clarify this?
Round 2
Reviewer 2 Report
The authors well answered my questions and addressed my concerns.